# Lessons learned from the COVID-19 pandemic about sample access for research in the UK

Jessica M Sims,[1] Emma Lawrence [ID],[1] Katy Glazer,[1] Amir Gander,[1] Barry Fuller,[1] Brian R Davidson,[1,2] Jonathan Garibaldi,[3] Philip R Quinlan [ID] [4]

[1]Division of Surgery and Interventional Science, University College London, London, UK
[2]HPB Surgery, Royal Free London NHS Foundation Trust, London, UK
[3]School of Computer Science, University of Nottingham, Nottingham, UK
[4]Digital Health, School of Medicine, University of Nottingham, Nottingham, UK

**Correspondence to**
Dr Philip R Quinlan;
philip.quinlan@nottingham.ac.uk

## ABSTRACT

**Objective** Annotated clinical samples taken from patients are a foundation of translational medical research and give mechanistic insight into drug trials. Prior research by the Tissue Directory and Coordination Centre (TDCC) indicated that researchers, particularly those in industry, face many barriers in accessing patient samples. The arrival of the COVID-19 pandemic to the UK produced an immediate and extreme shockwave, which impacted on the ability to undertake all crucial translational research. As a national coordination centre, the TDCC is tasked with improving efficiency in the biobanking sector. Thus, we took responsibility to identify and coordinate UK tissue sample collection organisations (biobanks) able to collect COVID-19-related samples for researchers between March and September 2020.

**Findings** Almost a third of UK biobanks were closed during the first wave of the UK COVID-19 pandemic. Of the remainder, 43% had limited capabilities while 26% maintained normal activity. Of the nationally prioritised COVID-19 interventional studies, just three of the five that responded to questioning were collecting human samples. Of the 41 requests for COVID-19 samples received by the TDCC, only four could be fulfilled due to a lack of UK coordinated strategy. Meanwhile, in the background there are numerous reports that sample collections in the UK remain largely underutilised.

**Conclusion** The response to a pandemic demands high level co-ordinated research responses to reduce mortality. Our study highlights the lack of efficiency and coordination between human sample collections and clinical trials across the UK. UK sample access is not working for researchers, clinicians or patients. A radical change is required in the strategy for sample collection and distribution to maximise this valuable resource of human-donated samples.

## BACKGROUND

Accessing well-annotated human samples for medical research is a crucial, yet complex process. Researchers face numerous barriers when it comes to accessing and using human samples, the ethical approval process is being one of them.[1] Research has also shown that industry end users face significant challenges in locating samples for research, resulting in the use of inferior alternatives such as animal models.[2] The current system of sample collection is still supported by research project funding that requires and encourages researchers to invest time and resource in the design, ethical and legal governance application processes and the acquisition of human samples. This is done in competition with other researchers, leading to an inescapable sense of 'ownership' rather than collaboration in the broader scientific endeavour. The United Kingdom Clinical Research Collaboration (UKCRC) Tissue Directory and Coordination Centre (TDCC) was established in 2014 to publicise existing sample collections and to improve sample access.[3] As of April 2020, the UKCRC TDCC's platform for sample discovery, the Tissue Directory, had 220 human biobanking organisations registered, including biobanks, cohort studies and clinical trials. Most of these organisations are hosted by public universities or National Health Service hospitals. Researchers can search this freely accessible, online directory to locate suitable research samples.[4] The TDCC seeks to improve efficiency in the sector by bringing visibility to existing infrastructure. The COVID-19 pandemic provided a challenge to biobanking for biomedical research but also an opportunity to gather data on failures and achievements to better integrate the sector in the future.

A recent survey of human tissue researchers demonstrated that it is still very common practice for researchers to collect samples by established local networks rather than sourcing and using samples located in other institutions.[1] Given the time and effort required to secure funding for research, the samples and associated data collected during any specific research programme are often perceived as an asset of the research group (academic or commercial) rather than a resource to be shared for the greater societal good. Therefore, despite the many mechanisms put in place by funders and regulators to promote transparency and reuse of existing infrastructures in research and

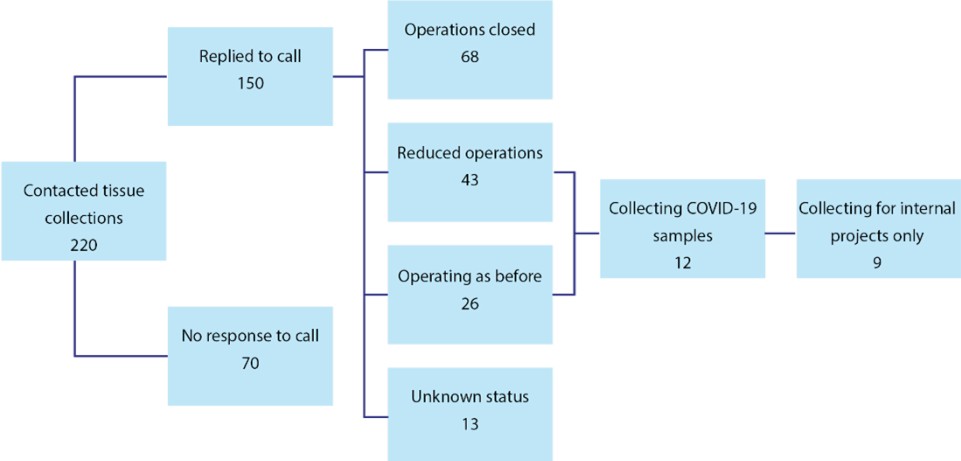

**Figure 1** Status of tissue banks in the UK during COVID-19 pandemic (March–September 2020).

dissemination of results, there are many understandable reasons and incentives why researchers would be reluctant to share samples widely.

Over a period of 25 weeks during the first wave of the COVID-19 pandemic in the UK (March to September), the TDCC received 41 requests for help to find COVID-19-related samples. The TDCC was able to assist just four researchers to source the samples they required.

### Human sample collection and sharing during the COVID-19 pandemic

All 220 tissue collection organisations registered with the UKCRC Tissue Directory were emailed up to three times enquiring about the state of their operations during the COVID-19 pandemic. Figure 1 summarises the flow of responses. One hundred and fifty tissue collection organisations responded (68%). From those, 111, or 74%, were closed (68) or had a reduced capacity to operate (43). Only 26 out of 150 organisations (17%) were operating as before the pandemic. The remaining 13 had an unknown operating status.

The closures or reduced capacity were categorised as due to:
► Site closure (17, 15.3%).
► Reallocation of staff or resources to support NHS COVID-19 efforts (17; 15.3%).
► Staff working from home or relocated (15; 13.5%).
► Suspension of non-essential research (10; 9.9%).
► Management or organisational decision (6; 5.4%).
► Study completed or not yet started (2; 1.8%).
► Health and safety concerns (1; 0.1%).
► No information given (43; 38.7%).

Of the 69 that were operating as before (26) or at reduced capacity (43), just 12 responded that they were able to access COVID-19 patients to collect COVID-19 samples. Furthermore, nine of these (75%) were already supplying samples to internal projects and were, thus, unable to share more widely to supply samples to additional external research projects.

It is striking that 74% of the UK human tissue collection organisations were either closed or not fully operational during this international health crisis. Nearly half of those that responded to our call were closed completely, rather than being seen to have skills and experience to benefit research or patient care. Also considering the high demand for aid to research into COVID-19, projects and organisations collecting COVID-19 samples missed an opportunity to support other research through publicising their capacity to collect or supply samples.

### DISCUSSION

This problem with sample sourcing impacted the ability of UK small to medium enterprises and public health institutions, and expert academic research groups to access samples for vital research projects.[5 6] For example, nearly two-thirds of the sample requests to the TDCC came from private research organisations. However, as the remaining third came from university research groups, sample access is a challenge faced by both public and commercial researchers. Current research culture (funders, regulators, rewards, precedent) is not configured to encourage or facilitate sharing, but rather it incentivises competition.

Substantial changes to the system will be required to offer a solution for improved human tissue access.[1 7] The COVID-19 pandemic has laid bare the consequences of a lack of a national strategy with proactive and upfront coordination and national infrastructure. We need a new approach for human sample research that will reassure patients that the scientific and biobanking communities are fully using their donated samples[8] in a manner, which is aligned with the ethos of consent given during donation.

### Contrast to coordinating and sharing of data

Each nation of the UK has support services to collect data for the NHS, which becomes a uniquely valuable resource to healthcare researchers on a scale that is unparalleled globally. NHS-X has recently been established to further the utilisation of data in leading data science initiatives. In stark contrast, there is no equivalent of NHS-D/X for tissue samples collected in the NHS.

Health Data Research UK (HDR UK) was established in 2018 and has now initiated many high-profile data coordinating efforts with a focus on open standards and interoperability, as the cornerstone of those activities. HDR UK could be considered, in some respects, an equivalent in data as TDCC is for patient samples. However, the most fundamental difference is that HDR UK has been able to fund theme-based data collection efforts in eight networks (BREATHE, DATA-CAN, Gut Reaction, INSIGHT, NHS DigiTrials, PIONEER, Discover-NOW and BHF Data Science Centre) and general capabilities under national research themes from their original dedicated and substantive sites (Cambridge, London, Midlands, North, Oxford, Scotland, South-West, Wales and Northern Ireland). HDR UK, therefore, has a ready-made funded network, in which it can direct and inform change, rather than simply advocating on the benefits of data sharing and collaborative working that has been the historical and disjointed case for human sample coordination efforts. System wide change is a challenge, HDR UK and the networks it coordinates, offers a beacon of light for a new way of operating across datasets.

The collection of human samples is in many ways a means to an end. Samples are the raw material from which many important clinical problems are investigated and addressed. High-quality data are the biproduct of a high-quality sample pipeline. TDCC has supported a data centric focus on biobanking, which places the emphasis on data sets with samples, rather than collections of 'samples with annotated data'. However, this has been an advocacy role that does not have regulatory or funding backing, which makes change hard to achieve using this strategy and relies on a collation of the willing.

In response to COVID-19, these data assets were brought to the fore and also further enabled by legislation.[9] £8.2M was invested in a new national data connectivity infrastructure under the National Core Studies programme for an initial period of 6 months that built on the established funding streams to support HDR UK and the national trusted research environments. This will result in legacy for future generations and data linkage and data infrastructure initiated in response to COVID-19 should easily become the new normal.

In contrast, there was no UK initiative to fund or drive the uplift of coordination efforts for human sample collection and sharing during the pandemic, and there was no regulatory change to support better use of samples in response to COVID-19. There were clearly national efforts with a research focus that had a direct or indirect aim of collecting samples, such as COVID-19 Genomics UK (COG-UK) Consortium, International Severe Acute Respiratory and emerging Infection Consortium (ISARIC) and Real-time Assessment of Community Transmission (REACT) study. However, once again there was no coordination between these studies. This research focus on sample collection, rather than sustaining operability, is the likely cause for so many tissue collection organisations closing during COVID-19. Instead, they could have been recognised as infrastructure with technical expertise, regardless of their research specialism, that could be repurposed as part of a national response. For example, if biobanks are so heavily defined by the research they can support, then it would be a logical consequence that while much other research was suspended, their operations would also be suspended. This had the result that the more service orientated biobanks that were still operational could not respond to the volume of demand than they then experienced.

A national approach to human tissue could bring the same efficiency and high-level governance that has been achieved with NHS data collection. The legal basis for tissue consent and its pathway (eg, registration, consent, opt-out, etc) could all be defined at a national level. The NHS stores millions of human tissue samples each year, which could be aligned with national data sets, for both academic and commercial research benefit. Tissue samples are vital to supporting research addressing and preventing diseases of today and tackling the pandemics of tomorrow.

## To the future

COVID-19 research has been strongly supported by the public. We have seen record participant recruitment in the Randomised Evaluation of COVID-19 Therapy Trial (RECOVERY trial),[10] public registering to be participants to test the new vaccines and symptom-alleviating medicines and mass public activities such as the ZOE app, which helped guide symptom advice. The enthusiasm and support from the public could be due to collective effort to support research to tackle an issue affecting everyone as well as timely information on how their participation is making a difference. The public are equally supportive of biobanking activities in other spheres, and consenting rates are always reported to be high.

Patients may understandably consider that tissue samples donated throughout their life and in altruistic fashion could be optimally used in multiple clinical trials or research studies through distribution from a suitably regulated biobank. However, currently their donated samples remain largely locked within individual studies and samples cannot contribute to national requirements as highlighted by the COVID-19 pandemic. The current approach of funding sample collections within research projects runs the risk that they become defined by that research project. The governance, the consent, the access and the focus are all defined by the study, rather than as a resource to be accessed and reused more widely towards the greater societal health benefit. The consequence of this approach of study led approach was clear to see in the response to COVID-19. There is an inseparable bond between the research themes which established the biobanks and what the biobank is able to support, meaning in response to COVID-19 many biobanks were closed. The national approach taken in COVID-19 repeated previous methods, which is to fund new collections of samples as part of research studies, with the consequent development of research silos where only retrospective collection was possible. To recognise the enormous value of human tissue in research, particularly when linked

with national data sets, we must consider new ways of coordination in sample collection activity if we truly seek a different research culture, in which samples are not preserved as property of a study but a donated gift from the public to further research.

As an example, what if sample coordination was embedded in relevant initiatives such as the clinical trials undertaken during COVID-19. With a different level of coordination, the RECOVERY trial could have taken a small aliquot of blood or consented for surplus blood to be collected alongside that taken for routine hospital tests. This then could have fulfilled sample access requests, like those received by TDCC and have supported a rapid and significant increase in UK research and development activity into COVID-19. This is of course not a criticism of RECOVERY, as they could only operate within the current constraints and expectations. However, aligning tissue or sample collection in parallel could have been achieved without added complexity or delays in the trial planning. In the bigger picture of the overall response to COVID-19, it was a missed opportunity. The challenge is that without a novel and coherent strategy, a coordinated approach for reuse is not seen as an opportunity, but as a radical and unworkable aspiration.

## CONCLUSION

The COVID-19 pandemic revealed that biobanking was not prioritised as an integral service, or infrastructure in its own right, to support crucial frontline research. This picture can be attributed to a wider lack of national strategy and coordination to facilitate access and efficient use of donated patient samples, which is not seen in allied areas of vital research imperative such as health data collection and analysis. Where time was of the essence at the start of the pandemic, wider sample demand was unmet, despite some facilities actively collecting samples for generic collections and specific studies—some with record participation in research. This indicates missed opportunities to maximise efficient use of resources and raises concern on how donated samples are used. A national strategy for biobanking, backed and reinforced by the governments, regulators and large research funders, would ensure a new level of coordination that regards routine sample collection as a national resource having the same value as routine data collection. This would then ensure that each donation could be fully used for active research rather than held as an asset for possible future use.

**Contributors** JS and EL contributed equally to this paper. JS, EL, KG, PRQ: substantial contributions to the conception or design of the work and the acquisition, analysis and interpretation of data for the work. They assisted in drafting the work and revising it critically for important intellectual content. All give final approval of the version to be published and agreement to be accountable for all aspects of the work in ensuring that questions related to the accuracy or integrity of any part of the work are appropriately investigated and resolved. BF, JG, BRD made substantial contributions to the analysis and interpretation of data for the work and revising it critically for important intellectual content. All give final approval of the version to be published and agreement to be accountable for all aspects of the work in ensuring that questions related to the accuracy or integrity of any part of the work are appropriately investigated and resolved.

**Funding** This work was supported by Medical Research Council grant number MR/R022488/1.

**Competing interests** None declared.

**Patient consent for publication** Not required.

**Ethics approval** Not applicable.

**Provenance and peer review** Not commissioned; externally peer reviewed.

**ORCID iDs**
Emma Lawrence http://orcid.org/0000-0001-9018-6010
Philip R Quinlan http://orcid.org/0000-0002-3012-6646

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
