## [Reviewer comments · BMJ Open]

ARTICLE DETAILS

TITLE (PROVISIONAL)	Lessons learned from the COVID-19 pandemic about sample access for research in the UK
AUTHORS	Sims, Jessica; Lawrence, Emma; Glazer, Katy; Gander, Amir; Fuller, Barry; Davidson, Brian; Garibaldi, Jonathan; Quinlan, Philip

VERSION 1 – REVIEW

REVIEWER	Lacerda, Eliana London School of Hygiene & Tropical Medicine, Clinical Research
REVIEW RETURNED	03-Sep-2021

GENERAL COMMENTS	This manuscript is a commentary, where the authors consider the current built capacity on biorepositories/biobanks in the UK and the current limitations to optimise their research capacity. Reflecting on the limited response from the existing human tissue samples collections to the Covid19 pandemic, the authors argue that a national strategy for biobanking is needed. This strategy should include a more efficient model for collecting and distributing samples to address the demands from academic and private researchers in a timely way, as it happens in the current pandemic. Considered the UK capacity for collecting and sharing data quickly, the authors argue a similar model should be applied for sample collections, even considering issues related to governance, consent, and sample distribution, if this effort is supported by the governments and large research funders. This is a well-reasoned manuscript, and my only suggestions are: 1. In page 5, line 58, the number should be 39, instead of 69 (I guess it is a typo).2. I would like to see some consideration in the manuscript, to the need of some specific biobanks/biorepositories, which require quite specific samples, as they should co-exist within a coordinated national strategy for sample collection that could be dynamic and highly responsive, if a well-thought strategy and appropriate research governance mechanisms are in place.
--

REVIEWER	Abdelhafiz, Ahmed Cairo University
REVIEW RETURNED	30-Sep-2021

GENERAL COMMENTS	Thank you for allowing me to review the revision of this manuscript. Biobanking industry has been affected to a variable degree by COVID-19 pandemic in different countries of the world. However, some lessons should have been learned, which could lead to improvement in the future, especially in aspects related to
--

	crisis management. The manuscript is interesting and well written. Kindly find my comments below. General comment: Please add the number of page at the bottom of each page of the manuscript. Specific comments  • Title: I am not sure if it will be understood from the title that this work is discussing sample access for research. I suggest adding the words "for research in UK" at the end of the title to clarify the goal of sample access and the country where these lessons were learned. • Background: page 5: it would be interesting to give more details about the biobanks mentioned in the background e.g. Are they public or private? Affiliated to hospitals or independent from them? Was the effect equal on both of them? • Page 5: What was the percentages of each factor of closures or reduced capacity of biobanks? • Discussion: To give the manuscript an international perspective, it would be useful if the authors provide some examples and lessons learned from other countries as well.
--	--

VERSION 1 – AUTHOR RESPONSE

Reviewer: 1

Dr. Eliana Lacerda, London School of Hygiene & Tropical Medicine

Comments to the Author:

This manuscript is a commentary, where the authors consider the current built capacity on biorepositories/biobanks in the UK and the current limitations to optimise their research capacity. Reflecting on the limited response from the existing human tissue samples collections to the Covid19 pandemic, the authors argue that a national strategy for biobanking is needed. This strategy should include a more efficient model for collecting and distributing samples to address the demands from academic and private researchers in a timely way, as it happens in the current pandemic. Considered the UK capacity for collecting and sharing data quickly, the authors argue a similar model should be applied for sample collections, even considering issues related to governance, consent, and sample distribution, if this effort is supported by the governments and large research funders. This is a well-reasoned manuscript, and my only suggestions are:

1. In page 5, line 58, the number should be 39, instead of 69 (I guess it is a typo).

This is not an error in the text. The sentence has been amended to clarify how we arrived at this number (69)

<< Of the 69 that were operating as before (26) or at reduced functioning in some capacity (43), just 12 responded that they were able to access COVID-19 patients to collect COVID-19 samples.>>

2. I would like to see some consideration in the manuscript, to the need of some specific biobanks/biorepositories, which require quite specific samples, as they should co-exist within a coordinated national strategy for sample collection that could be dynamic and highly responsive, if a well-thought strategy and appropriate research governance mechanisms are in place.

We argue that biobanks, regardless of specialism, could have responded to the pandemic if there was a coordinated national strategy that allowed them to do so. We have added a sentence in para 5 section "Contrast to coordinating and sharing of data"

Reviewer: 2

Dr. Ahmed Abdelhafiz, Cairo University

Comments to the Author:

Thank you for allowing me to review the revision of this manuscript. Biobanking industry has been affected to a variable degree by COVID-19 pandemic in different countries of the world. However, some lessons should have been learned, which could lead to improvement in the future, especially in aspects related to crisis management. The manuscript is interesting and well written. Kindly find my comments below.

General comment: Please add the number of page at the bottom of each page of the manuscript.

Page numbers have been added to the manuscript

Specific comments

Title: I am not sure if it will be understood from the title that this work is discussing sample access for research. I suggest adding the words "for research in UK" at the end of the title to clarify the goal of sample access and the country where these lessons were learned.

We have amended the title

Background: page 5: it would be interesting to give more details about the biobanks mentioned in the background e.g. Are they public or private? Affiliated to hospitals or independent from them? Was the effect equal on both of them?

We have added the sentence to para 1 Background: << Most of these tissue collection organisations are hosted by public universities or NHS hospitals.>>

In the UK, much high level translational research that requires sample access is done in close collaboration (often embedded in the same institutes) between NHS departments with academic expertise and university science groups. We have not added more information on the effect of the pandemic on university vs NHS hosted biobanks, as we did not ask for this information in our survey.

Page 5: What was the percentages of each factor of closures or reduced capacity of biobanks?

We have added this to the manuscript and is as follows:

<Site closure (17, 15.3%).

Reallocation of staff or resources to support NHS COVID-19 efforts (17; 15.3%).

Staff working from home or relocated (15; 13.5%).

Suspension of non-essential research (10; 9.9%).

Management or organisational decision (6; 5.4%).

Study completed or not yet started (2; 1.8%).

Health and safety concerns (1; 0.1%).

No information given (43; 38.7%).>>

Discussion: To give the manuscript an international perspective, it would be useful if the authors provide some examples and lessons learned from other countries as well.

The nature of TDCC oversight necessitates that this current analysis is UK-centric, but similar observations have been made on an international scale

“The Responses of Biobanks to COVID-19” Edited by Marianne K. Henderson, Zisis Kozlakidis, Authors: Jajah Fachiroh, Beatrice Wiafe Addai, Xun Xu, Sameera Ezzat, Heidi Wagner, Márcia M.C. Marques, and Birenda K. Yadav. *Biopreservation and Biobanking*. Vol: 18 Issue 6. Dec 2020.483-491. <http://doi.org/10.1089/bio.2020.29074.mkh>

“Opportunities and Risks for Research Biobanks in the COVID-19 Era and Beyond” Daniel Simeon-Dubach and Marianne K. Henderson. *Biopreservation and Biobanking*. Volume: 18 Issue 6. Dec 2020.503-510. <http://doi.org/10.1089/bio.2020.0079>